# Fine Structure Splitting of Phonon-Assisted Excitonic Transition in (PEA)_2_PbI_4_ Two-Dimensional Perovskites

**DOI:** 10.3390/nano13061119

**Published:** 2023-03-21

**Authors:** Katarzyna Posmyk, Mateusz Dyksik, Alessandro Surrente, Katarzyna Zalewska, Maciej Śmiertka, Ewelina Cybula, Watcharaphol Paritmongkol, William A. Tisdale, Paulina Plochocka, Michał Baranowski

**Affiliations:** 1Department of Experimental Physics, Faculty of Fundamental Problems of Technology, Wroclaw University of Science and Technology, 50-370 Wroclaw, Poland; 2Department of Chemical Engineering, Massachusetts Institute of Technology, Cambridge, MA 02139, USA; 3Laboratoire National des Champs Magnétiques Intenses, EMFL, CNRS UPR 3228, Université Toulouse, Université Toulouse 3, INSA-T, 31400 Toulouse, France

**Keywords:** 2D perovskites, exciton, phonon, optical spectroscopy, fine structure splitting, (PEA)2PbI4

## Abstract

Two-dimensional van der Waals materials exhibit particularly strong excitonic effects, which causes them to be an exceptionally interesting platform for the investigation of exciton physics. A notable example is the two-dimensional Ruddlesden–Popper perovskites, where quantum and dielectric confinement together with soft, polar, and low symmetry lattice create a unique background for electron and hole interaction. Here, with the use of polarization-resolved optical spectroscopy, we have demonstrated that the simultaneous presence of tightly bound excitons, together with strong exciton–phonon coupling, allows for observing the exciton fine structure splitting of the phonon-assisted transitions of two-dimensional perovskite (PEA)${}_{2}$PbI${}_{4}$, where PEA stands for phenylethylammonium. We demonstrate that the phonon-assisted sidebands characteristic for (PEA)${}_{2}$PbI${}_{4}$ are split and linearly polarized, mimicking the characteristics of the corresponding zero-phonon lines. Interestingly, the splitting of differently polarized phonon-assisted transitions can be different from that of the zero-phonon lines. We attribute this effect to the selective coupling of linearly polarized exciton states to non-degenerate phonon modes of different symmetries resulting from the low symmetry of (PEA)${}_{2}$PbI${}_{4}$ lattice.

## 1. Introduction

The two-dimensional (2D) Ruddlesden–Popper perovskites [1,2] bridge metal halide perovskites [3,4,5] with the 2D van der Waals crystals [6,7], combining in one semiconducting material system properties related to the soft ionic lattice [2,8,9] with highly pronounced excitonic effects [10,11,12]. Additionally, their good optical properties together with simple band structure engineering [1,2,10], means they are a fascinating platform for the investigation of excitonic and/or polaronic physics. 2D perovskites can be considered as natural quantum wells, consisting of octahedral units forming metal–halide slabs, separated by large organic spacers [1,2], which act as barriers, as we show in Figure 1a. The general formula describing 2D perovskites is A’${}_{2}$A${}_{n-1}$M${}_{n}$X${}_{3n+1}$, where A’ is a large monovalent organic cation acting as a spacer, A is a small monovalent cation (methylammonium—MA, formamidinium—FA, and cesium—Cs), M is a divalent cation that can be Pb${}^{2+}$ and Sn${}^{2+}$, X is an anion (Br${}^{-}$, I${}^{-}$), and n=1,2,3,… is the number of octahedral layers in the perovskite slab. For the thinnest form of 2D perovskites (n=1), the excitonic effects are extremely pronounced, due to quantum and dielectric confinement (see Figure 1a) [2,10,12]. The exciton binding energy can reach a few hundred of meV [10,11], which strongly enhances the exchange interaction and leads to a considerable fine structure splitting (FSS) of the exciton states [13,14,15,16,17,18,19,20,21,21].

The FSS of the band–edge exciton in 2D perovskite originates from electronic states, which, for both electrons and holes, are characterized by a total angular momentum 1/2 [13,14,15]. As shown in Figure 1b, there are four band–edge excitonic states characterized by a different total angular momentum (J=0 or 1) and by its *z*-component $J_{z}=0$ or ±1. The degeneracy between dark singlet (J=0) and bright triplet states (J=1) is always lifted by the exchange interaction and the dark state is the ground state [14,15,17,18,20]. Depending on the crystal symmetry, a partial or complete lifting of the bright exciton state degeneracy is expected [13,14,15,22]. In 2D materials, the symmetry is naturally broken in the *Z* direction, which splits off the state with the dipole moment oriented along the out-of-plane direction (*Z*-state). If the in-plane symmetry of the crystal is low, the degeneracy between the two remaining bright states is further lifted, leading to the exciton state ladder as shown in Figure 1b. Two split and linearly polarized *X* and *Y* states with dipole moments oriented in the plane of the crystal can be easily observed in the photoluminescence (PL) and reflectance (R) spectra in archetypical 2D perovskites (PEA)${}_{2}$PbI${}_{4}$ (where PEA stands for phenylethylammonium) as reported recently [16,17,19] and shown also here in panel (c) of Figure 1.

The tightly bound exciton in 2D perovskites facilitates exciton states’ splitting in the range of a few up to a few tens of meV [17,18,23,24], which is orders of magnitude larger than in other semiconductor nanostructures [25,26]. Simultaneously, the excitons in 2D perovskites are strongly coupled to lattice vibrations [2,27,28,29]. The interplay between enhanced exciton FSS and polaronic effects is believed to be an origin of rich emission [9,16,17,18,19,23,24] and absorption [15,21,22,27,28,29,30,31] spectra with multiple characteristic sidebands, whose detailed understanding still remains elusive and is the subject of ongoing debate [27,28,29,32].

Here, we report on another fascinating aspect of the optical response of 2D perovskites. For the first time, we show that the characteristic for (PEA)${}_{2}$PbI${}_{4}$ 2D perovskite multiple sidebands are linearly polarized and exhibit energy splitting as the main excitonic transitions. We propose that the concomitant presence of a significant FSS together with strong exciton–phonon coupling enables observing the FSS of the sidebands that we attribute with the phonon-assisted transitions (phonon replica). Interestingly, we notice that the splitting of these transitions can be different from the one characteristic of the zero-phonon lines. This suggests a selective coupling of the split in plane excitonic states to different phonon modes (with different energies). Our results highlight the importance of the internal lattice symmetry of 2D perovskites, which lifts the degeneracy of the exciton state and phonon modes, resulting in an intriguing optical response.

## 2. Materials and Methods

The investigated crystal was synthesized with the use of cooling-induced crystallization [33,34]. A mixture of lead(II) oxide (PbO, 0.558 g, and 2.5 mmol), phenethylamine (198 μL, 2 mmol), and hypophosphorus acid (H${}_{3}$PO${}_{2}$, and 425 μL) was dissolved in 8 mL of 55% hydrogen iodide solution (HI) to form a bright yellow solution at 130 ${}^{\circ }$C. After that, the solution was allowed to cool slowly to room temperature to yield (PEA)_2_PbI${}_{4}$ crystals.

For the PL and reflectance measurements, the sample was mounted on the cold finger of a He-flow optical cryostat. All the measurements were performed at 4.2 K. The PL was excited with a 405 nm CW laser. For the reflectance measurements, the white light was provided by a broad-band halogen high-intensity fiber light source. The excitation and signal collection were performed with the use of a long working distance, 50× magnification microscope objective with a numerical aperture of 0.55. The signal was analyzed using a 50 cm long monochromator, equipped with a grating of 1200 grooves per mm and detected using a liquid-nitrogen-cooled CCD camera (providing spectral resolution of 0.1 nm). The polarization-resolved spectra were obtained with by using a linear polarizer (Glan crystal) and a broadband half-wave plate.

## 3. Results and Discussion

Figure 1c shows the unpolarized reflectance and PL spectra of the investigated (PEA)${}_{2}$PbI${}_{4}$ crystal. Two resonances at around 2.352–2.355 eV are resolved in the reflectance spectrum. They correspond to the two in-plane bright exciton states [17], named X and Y here. The presence of these two free excitonic transitions can be also noticed in the PL spectrum as two weak peaks on the high-energy slope of the dominating PL peak [17]. The detailed origin of the main PL band is not well established. However, there are indications that it could be linked to the recombination of the exciton localized by local lattice distortions known as exciton–polarons, i.e., exciton states “dressed” with a local lattice reconfiguration resulting from the soft and the ionic nature of perovskites [9,17].

New interesting features of the R spectrum are observed in the broader spectral range, as presented in Figure 2a (black line). The two in-plane exciton states are followed by two broad side bands, SB${}_{1}$ and SB${}_{2}$, on the high-energy side. Their presence becomes more evident when we calculate the derivative of the reflectance spectrum, shown as a red curve in Figure 2a, whose minima correspond to the transition energies [35]. These two transitions are blue-shifted by around ∼25 meV and ∼45 meV from the sharp excitonic resonances identified as the X and Y exciton states.

To interpret the origin of the two sidebands, we refer to Franck–Condon’s picture [36,37] presented schematically in Figure 2b,c. When an electronic excitation results in a local lattice reconfiguration, the transition probability is modulated by the overlap between the vibrational lattice levels in the ground and the excited state of the lattice (see the upper and lower parabola in Figure 2b). The absorption and emission spectrum consist of a series of transitions separated by the energy of a phonon (series of phonon replicas). The difference between the equilibrium position of atoms in the ground ($Q_{g}$) and the excited lattice state ($Q_{e}$) determines the average energy $E_{R}$ that is emitted by the lattice to reach the equilibrium position upon excitation or recombination. This corresponds to the average number of phonons emitted, referred to as the Huang–Rhys factor [36,37]. As the $Q_{e}-Q_{g}$ increases, the absorption and emission spectrum change from being dominated by the zero-phonon line to absorption and PL lines assisted by multiple phonon emission. This results in the Stokes shift between absorption and PL spectra. These are blue- and red-shifted from the zero-phonon line by $E_{R}$, respectively, and form a mirror image, as schematically presented in Figure 2c [36,37]. Usually, for a significant electron–phonon coupling, the phonon-assisted absorption or emission lines merge into a broad band—phonon wing—shown as envelope lines in Figure 2c).

Therefore, we ascribe each of the sidebands to the merged responses of different order phonon replicas (different numbers of phonons participating in the transition), i.e., phonon wings [36], where SB${}_{1}$ involve phonons of different energies than SB${}_{2}$. Similar side band features observed in the transmission or PL measurements of (PEA)${}_{2}$PbI${}_{4}$ were also previously ascribed to phonon replicas [22,27,28,30]. A more detailed description of the complex 2D perovskite optical spectra still remains elusive and is a subject of ongoing discussion [27,28,29,32]. Nevertheless, as we further demonstrate, some aspects of this high-energy spectrum can be well captured within Franck–Condon’s picture, which can explain its peculiar polarization-resolved response discussed in the next part.

As it is shown in Figure 3a, by choosing a particular reflected light polarization direction, we probe one of the in-plane exciton states. For instance, for the two selected orthogonal polarizations (called $\pi_{X}$ and $\pi_{Y}$), we can exclusively probe the X state (red curve) at ∼2.353 eV or the Y state at ∼2.355 eV. They are separated by ∼2.1 meV, which is in agreement with previous reports [16,17,19]. Figure 3b shows the dependence of the reflectance spectrum in the energy range corresponding to the X and Y excitons versus the detection polarization angle. This dependence confirms the linear polarization of the X and Y transitions. The energy of the exciton resonance exhibits a sinusoidal behaviour, which is characteristic of two transitions split in energy and with dipole moments oriented orthogonally. This trend can be very nicely reproduced by the $\Delta_{XY}\sin^{2}\left(\alpha \right)$ curve plotted as a dashed black line. $\Delta_{XY}=2.1$ meV corresponds to the in-plane bright exciton FSS and α is the detection angle.

SB${}_{1}$ and SB${}_{2}$ are also linearly polarized as shown in Figure 3a,c,d. The energy position as a function of the detection polarization angle also exhibits an oscillatory trend, as presented in Figure 3c,d. The phase of these oscillations is exactly the same as for the X and Y exciton states. They maintain the selection rules of the in-plane exciton states, which is a strong indication that both sidebands derive from the X and Y states. Interestingly, the splitting observed for SB${}_{1}$ is the same as for the X and Y states, while the $\Delta_{XY}$ for SB${}_{2}$ is around 2 meV larger (see also dashed line in Figure 3d). This particular behaviour of the sidebands can be captured by a simple Franck–Condon model with the assumption of selective coupling of the split excitonic states to different phonon modes.

Under the assumption that the sidebands visible in the reflectance spectrum are phonon wings that maintain the selection rules of zero-phonon lines in the $\pi_{X}$ polarization, we observe the X exciton with its two sidebands, and, in the $\pi_{Y}$, we observe the Y exciton and the two corresponding phonon wings. The splitting of the sidebands for the two orthogonal polarizations can be related to two factors: (i) an inherent splitting of the X and Y exciton states and (ii) a selective coupling of the X and Y transitions to phonons with a different energy. It is straightforward to show that the splitting of the sideband transitions ($\Delta_{SB}$) visible in reflectance for $\pi_{X}$ and $\pi_{Y}$ polarization should be
(1)$$\Delta_{SB}=\Delta_{XY}+\left(E_{R}^{Y}-E_{R}^{X}\right)$$
where $E_{R}^{X/Y}$ is the average energy emitted by the lattice upon excitation of the X or Y exciton. The splitting observed for the $\pi_{X}$ and $\pi_{Y}$ polarization of SB${}_{1}$ is exactly the same as for the X and Y transitions, which means that $E_{R}^{X}=E_{R}^{Y}$ and phonons, forming SB${}_{1}$, coupled to X and Y transitions have the same energy. However, for SB${}_{2}$, $\Delta_{XY}$ is around 4 meV, which means that the $E_{R}^{X}\neq E_{R}^{Y}$ and $E_{R}^{Y}-E_{R}^{X}\approx 2$ meV. Therefore, in this case, the phonons coupled to different in-plane exciton states have a slightly different energies, which results in the enhanced splitting of the SB${}_{2}$ transitions visible in the reflectance spectrum. To support this interpretation, we also analyze the PL response. In the PL spectra presented in Figure 4, we note an emission peak red-shifted from the zero-phonon lines of X and Y states by around 45 meV and 47 meV, respectively, which is the same as SB${}_{2}$. Therefore, we ascribe this emission to a PL phonon wing corresponding to the SB${}_{2}$ visible in the reflectance. The splitting of this PL peak measured in the $\pi_{X}$ and $\pi_{Y}$ polarization is negligible. However, in the PL response, the splitting of the phonon wing should be decreased by the $E_{R}^{Y}-E_{R}^{X}$: (2)$$\Delta_{SB}^{PL}=\Delta_{XY}-\left(E_{R}^{Y}-E_{R}^{X}\right).$$
Thus, since $E_{R}^{Y}-E_{R}^{X}\approx \Delta_{XY}$, we observe a vanishing splitting of the PL peak corresponding to SB${}_{2}$, which corroborates our interpretation of the selective coupling of the excitonic states to the phonon modes with slightly different energy. It is worth noting that the low symmetry of the lattice of (PEA)${}_{2}$PbI${}_{4}$ [38] facilitates both the lifting of the degeneracy of exciton states [13,14,15] and that of some of the phonon modes [28,39]. Thus, assuming that excitons with different dipole orientation selectively couple to lattice vibration with given symmetries [28,39,40], we can expect different splittings of this state than observed for the zero-phonon line. Moreover, the same mechanism can lead to a contrasting splitting of the phonon-assisted transitions observed in the absorption and emission process.

## 4. Conclusions

We have shown that the high-energy sidebands visible in the reflectance spectrum of (PEA)${}_{2}$PbI${}_{4}$ are linearly polarized and exhibit energy splitting between the two orthogonal polarized states. Our analysis has suggested that the observed effect can be understood as an interplay between the exciton fine structure splitting and the exciton–phonon coupling. Interestingly, we have noticed that the splitting of the phonon wing bands can be different than the one characterizing the zero-phonon line, which points to the selective coupling of the differently oriented excitonic states to non-degenerate phonon modes. This highlights the importance of the internal lattice symmetry of the 2D perovskites, as it lifts the degeneracy of the phonon modes and exciton states. We support our interpretation with the use of the simple Franck–Condon model, which also provides an explanation for the contrasting splitting of the transition in the PL and the reflectance spectra for the phonon-assisted transitions. We expect that the presented result can help to construct more detailed models of excitonic species, thus leading to a comprehensive understanding of the optical properties of 2D perovskites with benefits for their applications.

## Figures and Tables

**Figure 1 nanomaterials-13-01119-f001:**
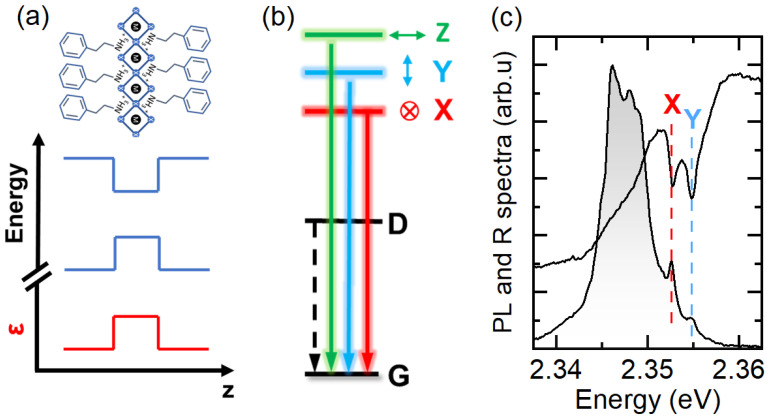
(**a**) Schematic view of a 2D perovskite of thickness n=1 together with the conduction and valance band alignment and spatial dependence of dielectric screening. (**b**) Schematic of the band edge exciton fine structure of (PEA)${}_{2}$PbI${}_{4}$ (PEA—phenylethylammonium). G is the ground state of the system (no excitons), D is the dark state, and X, Y, and Z are the three bright states with orthogonally oriented dipole moments. (**c**) Photoluminescence (grey shading) and reflectance (black line) spectra of (PEA)${}_{2}$PbI${}_{4}$ single crystal. The two in-plane bright exciton states are indicated by X and Y.

**Figure 2 nanomaterials-13-01119-f002:**
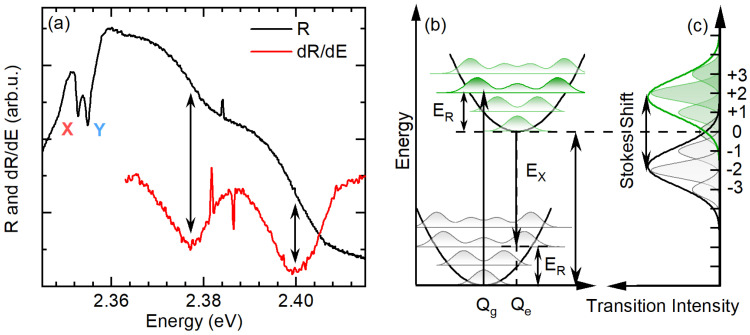
(**a**) Broad spectral range reflectance spectrum of (PEA)${}_{2}$PbI${}_{4}$ (black line) and its derivative (red line). The minima in derivative correspond to transition energies. (**b**) Schematic showing the Franck–Condon model (configurational coordinate diagram). The lower and upper parabolas represent the harmonic potential for the lattice in the ground and excited state. The shaded area corresponds to the spatial probability distribution of the harmonic oscillator. The arrows show examples of the transition from the ground (excited) vibrational level 0 to the vibration level 2 of the excited (ground) state. Such a transition results in the emission of two phonons. (**c**) Schematic of absorption (green) and emission (grey) spectra of exciton coupled to lattice vibration. The envelope lines represent the merged responses of phonon replicas involving different numbers of phonons also known as phonon wings.

**Figure 3 nanomaterials-13-01119-f003:**
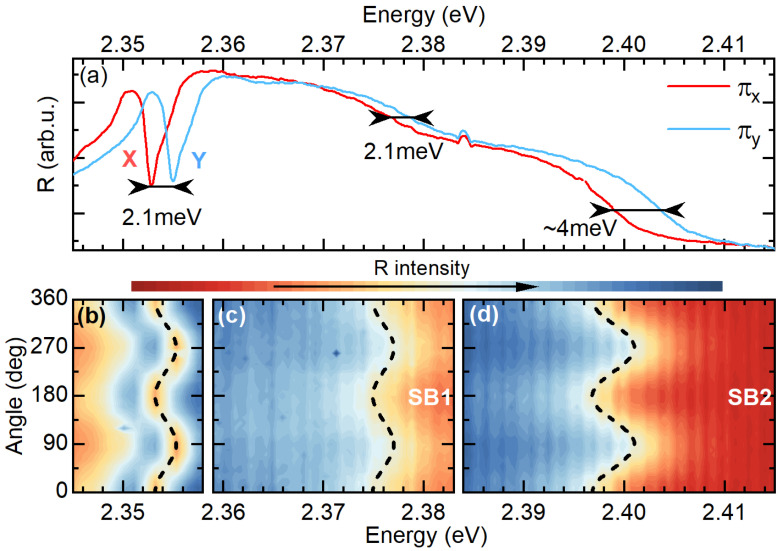
(**a**) Polarization resolved spectra of reflectance for $\pi_{X}$ and $\pi_{Y}$ polarization. (**b**–**d**) Dependence of the reflectance spectrum as a function of the polarization angle. The intensity in each panel is adjusted to emphasize the splitting in a given spectral range. $\Delta_{XY}\sin^{2}\left(\alpha \right)$ is plotted in dashed black lines, which shows the oscillatory behaviour of the transition energy as a function of detection angle.

**Figure 4 nanomaterials-13-01119-f004:**
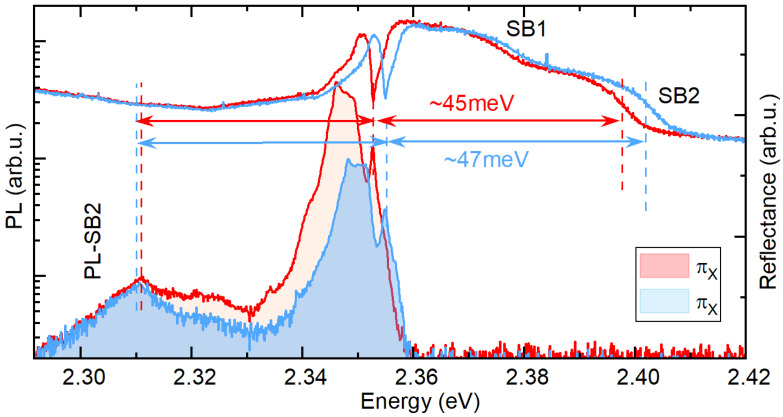
Polarization resolved PL and reflectance spectra. Blue and red arrows show equal distances of low energy PL peak and SB2 from the zero-phonon line of X and Y exciton.

## Data Availability

The data presented in this study are available on request from the corresponding author.

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
