# Peer review of "Fine Structure Splitting of Phonon-Assisted Excitonic Transition in (PEA)_2_PbI_4_ Two-Dimensional Perovskites"

_nanomaterials, 2023, doi:10.3390/nano13061119_

Round 1
Reviewer 1 Report
The manuscript reports fine structure splitting of phonon-assisted excitonic transition behavior in (PEA)2PbI4 2D Perovskites. The paper is well written, and the results are of interest for the scientific community working on 2D perovskites. I recommend its publication can be accept in present form.
Author Response
We would like to thank the referee for his very positive feedback.
Reviewer 2 Report
The work is devoted to spectroscopic studies of the 2D perovskite (PEA)2PbI4. The exciton fine structure splitting of the phonon-assisted
transitions is observed.
The results are very interesting, promising and have to be published.
Just a few recommendations on how to improve the manuscript presentability
1) Please decipher all the abbreviations at the first use. For example, PEA is deciphered only in line 45, when it occurs for the forth or fifth time; FA and MA (line 25) are not deciphered at all (and I can only guess that they stand for formamidinium and methylammonium), etc.
The same goes for drawings. The authors explain the meaning of D,X,Y,Z in Fig. 1b but what is 'G'? (of course, I understand what it is but it has to be explained in figure caption)
Why do the authors use all sorts of options: 'zero-phonon line', '0-phonon line' and even '0 phonon line'?
2) Please check all the chemical formulas. For some reason (PEA)2PbI4 turned into PEA2PbI4 at some point and never came back. The formula in line 24 does not describe a chemical compound, but a charged radical (there must be X3n+1 instead of Xn+1), etc.
3) Does it make sense to move the brief explanation of the Franck–Condon model (along with Fig. 3a) to an earlier part of the article?
4) Conclusion section contains no information about where such interesting phenomena can be applied.
Author Response
We would like to thank the referee for his positive opinion of our work and valuable comments. In the attachment, we present responses to the referee's questions.

Reviewer 3 Report
Dear Authors,
Your work is interesting, but some revisions are required as follows.
1. It is recommended to not introduce formulas or abbreviations in the title or in text without explaining them. See PEA, for example. Check the entire text for such problems.
2. Please specify if Fig. 1 is original or from references, in the second case you must insert the proper references.
3. Introduction: “The general formula describing 2D perovskites is A’2An−1MnXn+1, where A’ is a large monovalent organic cation acting as a spacer, A is a small monovalent cation (MA, FA, Cs) …” - what are MA and FA? Please explain in text.
4. Regarding novelty: please clearly point out the novelty of your work comparatively to already published papers, in the Introduction chapter. Is this the first time this effect presented? Is this compound for the first time analysed?
5. If possible insert some apparatus errors in Materials and Methods chapter.
6. Please insert X and Y in Fig. 2 too, not only in Fig. 1 c).
7. Please insert the same formula for (PEA)2PbI4 in all text. See PEA2PbI4 at pages 3, 4 and 6 of the PDF file.
8. Do not insert any equations which are not original without References. See equations (1) and (2).
9. Fig. 3 (a): is this figure original? If not, insert the appropriate reference.
10. Page 5 of the PDF file: “Blue and red arrows show equal distances of low energy PL peak and SB2 from the 0-phonon line of X and Y exciton.
replicas). The difference between the equilibrium position of atoms in the ground (Qg) and the excited lattice state (Qe) determines the average energy ER that is emitted by the lattice to reach the equilibrium position upon excitation or recombination.” – as seen it has no meaning, probably it must be there a line or more missing in text. Didn’t you check the manuscript before sending? Be more careful to such problems!
11. Do not insert models or factors name without appropriate reference: see “Huang-Rhys factor” at page 5 of the PDF file.
12. All discussion and assumptions about the splitting, at pages 5-6 of the PDF file, must be proved by other authors works or by other investigations, otherwise it seams rather speculative. Please discuss this.
Author Response
We would like to thank the referee for his overall positive feedback and valuable comments. In the attachment, we present detailed answers to the referee's questions.
